

# Self-reported depression and anxiety rates among females with cutaneous leishmaniasis in Hubuna, Saudi Arabia

Nahid Elfaki[1], Mohammed Alzahrani[2], Yahya Hussein Ahmed Abdalla[1], Mugahed Ali Alkhadher[3], Abdalla MohamedAhmed Osman[1], Wargaa Taha[4], Wael Alghamdi[5], Faroq Abdulghani Alshameri[6] and Waled A. M. Ahmed[5]

[1] Community Health Nursing Department, College of Nursing, Najran University, Najran, Saudi Arabia
[2] Pediatric Department, College of Medicine, Najran University, Najran, Saudi Arabia
[3] Medical and Surgical Nursing Department, College of Nursing, Najran University, Najran, Saudi Arabia
[4] Maternity and Child Health Nursing Department, College of Nursing, Najran University, Najran, Saudi Arabia
[5] Nursing Department, Faculty of Applied Medical Sciences, Al-Baha University, Al-Baha, Saudi Arabia
[6] Nursing Department/Faculty of Medical Sciences and Nursing, Al-Rayan College, Almadina, Saudi Arabia

Corresponding authors
Yahya Hussein Ahmed Abdalla,
yabdalla286@gmail.com
Waled A. M. Ahmed,
wahmed@bu.edu.sa

## ABSTRACT

**Background:** Cutaneous leishmaniasis (CL) is a parasitic disease transmitted through the bite of infected sandflies, causing disfiguring skin lesions and a range of physical symptoms. However, the psychological impact of CL is often overlooked despite the significant burden it imposes on the affected individuals and communities. This is especially true in Saudi Arabia, where limited research exists on the psychological consequences of CL, particularly among females. This study aimed to address this knowledge gap by investigating the perceived psychological problems associated with CL among females living in the Hubuna area of Saudi Arabia.

**Methods:** This cross-sectional study recruited 213 females with CL in the Hubuna area of Saudi Arabia using purposive sampling. Data was collected using a self-administered electronic questionnaire that included socio-demographic characteristics and measures of depression and anxiety using the BDI and GAD-7 tools. Descriptive analysis was used to determine the psychological impact of CL, including means and standard deviations for the BDI and GAD-7 scores, as well as frequencies and percentages for other variables of interest. Logistic regression was performed to identify independent predictors of anxiety and depression, including variables such as age, marital status, education, occupation, number and location of lesions. The significance level for all statistical tests was set at $p < 0.05$. The study was carried out between September and December of 2022.

**Results:** The study found that the mean Beck Depression Inventory (BDI) and Generalized Anxiety Disorder-7 (GAD-7) scores among the participants were 8.67 ± 4.82 out of 63 and 8.20 ± 7.08 out of 21, respectively. Furthermore, the prevalence of depression and anxiety was 55.9% and 68.1%, respectively, indicating a significant psychological burden associated with CL in the study population. The results of the logistic regression analysis showed that anxiety and depression were significantly

associated with age, marital status, number of lesions, and location of the lesions on the body, highlighting the importance of considering these factors when designing interventions aimed at improving the mental health of CL patients.

**Conclusions:** In conclusion, this study highlights the significant psychological impact of CL among females in the Hubuna area of Saudi Arabia, calling for urgent action to address this neglected aspect of the disease. By integrating mental health considerations into CL prevention and management efforts, healthcare providers can improve the overall well-being of affected individuals and contribute to the broader goal of eliminating CL as a public health concern.

## INTRODUCTION

Cutaneous leishmaniasis (CL) is a skin disease caused by parasites called *Leishmania major* and *Leishmania tropica* (*Chaouch et al., 2019*). CL is a serious dermatological disease that primarily affects developing societies worldwide (*Bilgic-Temel, Murrell & Uzun, 2019*; *World Health Organization, 2008*). CL is usually transmitted to humans through the bite of infected female sandflies, which are typically active at dusk and at night in tropical and subtropical regions around the world, including parts of Asia, Africa, the Middle East, and Latin America (*Heirwegh et al., 2021*; *El-Mouhdi, Chahlaoui & FekhaouI, 2020*).

CL can cause either a single or multiple large, locally-destructive skin lesions that may result in anxiety, depression, and self or social stigma (*Kumosani et al., 2022*; *Alzahrani et al., 2023*; *Abuzaid et al., 2017*). Another problem is that women and children are especially vulnerable to the psychological consequences of this infection, as it causes disfiguring skin lesions that can leave lifelong scars (*Bennis et al., 2018*; *Yanik et al., 2004*; *Kassi et al., 2008*). Additionally, some women with CL present late to health services with severe unwanted complications, such as large ulcerative crusted nodules, nasal deformities, disfigured lips, large scars, or even malignant transformations. (*El-Mouhdi, Chahlaoui & FekhaouI, 2020*; *Bettaieb et al., 2020*; *Bennis et al., 2018*; *Kumosani et al., 2022*; *Turan et al., 2015*; *Sirey et al., 2001*).

CL social stigma (CLSS) significantly impacts the lives of affected individuals, including their social interactions, marriage prospects, and the ability to find employment. Unfortunately, women with CL are particularly victimized as they are considered unacceptable for marriage and are sometimes no longer accepted by their own families (*Bennis et al., 2018*; *Yanik et al., 2004*; *Kassi et al., 2008*).

Many research studies have shown that the stigma and discrimination associated with CL can negatively impact emotional well-being and mental health. People living with CL may start to see themselves in a negative light due to the stigma they experience, which can lead to feelings of anxiety and depression. Furthermore, they may also be afraid of discrimination or judgment if others find out about their condition (*Reithinger et al., 2005*; *Bennis et al., 2018*; *Ramdas, van der Geest & Schallig, 2016*).

Research conducted in various countries, including Yemen (border country to Saudi Arabia), Afghanistan, Tunisia, and Morocco have demonstrated that young women who are affected by may face significant psychological challenges. These challenges include difficulties in finding a suitable marriage partner, or may be prohibited from living with their partners (*Al-Kamel, 2016a*; *Reithinger et al., 2005*; *Chahed et al., 2016*; *Bennis et al., 2017*, *2018*).

Specifically, in Saudi Arabia, CL is considered a major public health problem due to its high prevalence rates and the significant associated impact on the psychological wellbeing and social life of affected individuals (*Kumosani et al., 2022*; *Alzahrani et al., 2023*; *Abuzaid et al., 2017*). Therefore, this survey aims to explore potential anxiety and depression rates due to CL in Saudi Arabia to highlight the impact of CL on female mental health and help reform subsequent intervention strategies and services addressing this problem.

# MATERIALS AND METHODS

## Study area and population

The current study was conducted in the Hubuna area, which is situated in the governorate of Najran in the southern part of Saudi Arabia. Hubuna is specifically located along the border with Yemen. It is worth noting that the total population of Hubuna was determined to be 20,400 individuals according to the most recent census conducted in 2017 by the Saudi General Authority for Statistics (SGAS) (*General Authority for Statistics, 2017*). Within the study area of Hubuna, cutaneous leishmaniasis (CL) is considered endemic. This means that the disease is regularly found within the population of the region. In recent years, there have been reported outbreaks of CL in Hubuna, indicating a heightened occurrence of the disease during those periods (*Alzahrani et al., 2023*).

## Study design

This was a cross-sectional study that took place between September and December of 2022.

### Inclusion criteria

The study included females 10 years of age or older who were currently living in Hubuna and had experienced cutaneous leishmaniasis (CL) infection, regardless of other demographic factors.

### Exclusion criteria

Females aged less than ten years or who did not have a history of CL infection and those who declined to participate in the study were not included in the sample. Additionally, patients who were severely ill, had low cognitive function, or were disoriented were excluded from the study as they may not have been able to accurately answer the questionnaire.

## Sampling and sample size

A purposive sampling technique was used to select the sample and determine the sample size based on the feasibility and practicality of conducting the study within the given resources and time frame. The minimum necessary sample size was obtained.
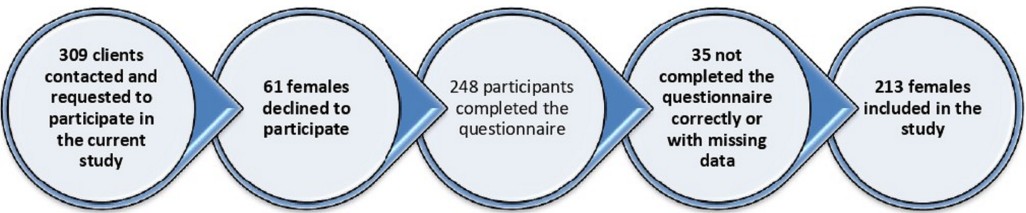

**Figure 1 The participants' recruitment flowchart.** This figure depicts the flowchart of participant recruitment for the study. The recruitment process began with initial screening, followed by eligibility criteria checks.

Assumptions related to the expected prevalence of depression and anxiety among females with cutaneous leishmaniasis in the study area, as well as the anticipated effect size, were used to calculate the minimum required sample size.

The purposive sampling technique involves selecting individuals who meet specific criteria that are relevant to the research question, such as age, gender, or health status. This approach can help ensure that the sample is representative of the population being studied, although it may introduce some bias if the criteria used to select participants are not appropriate. Overall, the use of purposive sampling in this study allowed the researchers to target specific groups of females who were more likely to be affected by cutaneous leishmaniasis and therefore obtain more accurate data on the mental health impact of the disease.

A total of 248 individuals completed the survey out of the 309 individuals who were contacted, resulting in a response rate of 80.3%. After removing incomplete questionnaires, the final sample size used for analysis was 213 participants, as shown in Fig. 1.

## Data collection process

After consenting to the study, participants were provided with a link to an online questionnaire *via* social media platforms (such as WhatsApp and Facebook). The link was only shared with eligible participants who met the inclusion criteria. The questionnaire was designed to assess their perception of CL and its impact on their mental health.

The questionnaire was composed of three main sections: section one was for gathering socio-demographic characteristics data; section two assessed the level of anxiety associated with CL ulcers (skin lesions) using the Generalized Anxiety Disorder-7 item scoring system (GAD-7) (*Plummer et al., 2016*); and section three assessed the level of depression associated with CL lesions using the Beck Depression Inventory (BDI) system.

Anxiety was measured using GAD-7, which is a self-report screening tool with seven questions about psychological symptoms. Participants were asked how often during the disease course they were bothered by each of the seven symptoms using a 4-point Likert scale system (from 0–3) where a zero indicated the symptom did not bother them at all or the question was irrelevant and a three indicated the participant was bothered by that symptom nearly every day. The total score ranged from 0–21, with higher scores indicating more anxiety symptoms. In this study, Cronbach's alpha coefficient of GAD-7 was 0.865.

Depression was measured using the Beck Depression Inventory (BDI) system. BDI is a self-assessment scoring system tool composed of 21 questions that are answered using the same 4-point Likert scoring system (0–3). A zero indicated a normal response with no sign of depression, one indicated a mild depressive response, a two indicated a moderate depressive response, and a three indicated a severe depressive response. Patients were asked to select the number that best represented their condition. The final BDI score for each participant was calculated by adding the total of all question responses, with the final score ranging from zero to a maximum of 63. The higher the score, the greater impact of the disease on the psychological health of the patient (*Wang & Gorenstein, 2013*).

The questionnaire was translated into Arabic and tested for content validity by five medical experts. The Cronbach's alpha test showed an acceptable reliability score of 0.82. A pilot study was then performed with 10 participants who were excluded from the main sample to verify the questionnaire's clarity and applicability, with changes made to the final questionnaire based on the pilot study results. All study participants whose BDI/GAD-7 scores indicated severe depression or anxiety were advised to see a psychologist for further evaluation.

## Statistical analysis

After cleaning and coding the data, the data analysis was performed using SPSS version 23 (IBM Corp., Armonk, N. Y, USA). Descriptive statistics were used to display the demographic characteristics of the participants. Data were reported as mean ± SD, and categorical data were reported as percentages. Categorical variables, such as gender, educational level, and age groups, were summarized and reported in terms of frequency distribution. Bivariate associations were used to explore the relationships between continuous variables. In addition, a logistic regression analysis was conducted to estimate odds ratios (OR) with 95% confidence intervals (CI) for anxiety symptomatology. A $p$-value of <0.05 was considered statistically significant.

## RESULTS

A total of 213 female participants were included in this analysis with the following characteristics and findings:

## Socio-demographic characteristics of the studied sample (*n* = 213)

Table 1 shows that the majority of study participants were between 21–40 years old (72.7%), with a smaller percentage falling within the 10–20 age range and their mean age was 26.4 ± 14.8 years. More than half of the participants were classified as singles (54.5%), while 28.6% were married, and 16.9% were separated, divorced, or widowed.

Nearly half of the patients (48.4%) attended intermediate or secondary school, while 32.4% were university graduates and 19.2% had no education or had only completed primary school. The majority of patients (68.1%) lived on or near farms or houses within plant areas. Only 36.2% of participants were employed, as most (63.8%) participants were children, students, or unemployed.

**Table 1 Socio-demographic characteristics of participants (n = 213).**

| Variable | Characteristics | Frequency | Percent (%) |
|---|---|---|---|
| Age in years | 10–20 | 23 | 10.8 |
| | 21–30 | 81 | 38 |
| | 31–40 | 74 | 34.7 |
| | ≥41 | 35 | 16.4 |
| Marital status | Single (did not experience marriage) | 116 | 54.5 |
| | Single (separated, divorced, widow) | 36 | 16.9 |
| | Married | 61 | 28.6 |
| Level of education | Uneducated or primary school | 41 | 19.2 |
| | Intermediate + secondary schools | 103 | 48.4 |
| | Higher education (University Graduates) | 69 | 32.4 |
| Living environment | Inside or nearby farms/multi-planted houses | 145 | 68.1 |
| | Far from farms/no plants inside houses | 68 | 31.9 |
| Employment status | Children/students/no jobs | 136 | 63.8 |
| | Employed female | 77 | 36.2 |
| Location of lesions | Head, face, and neck | 119 | 55.9 |
| | Upper extremities | 43 | 20.2 |
| | Lower extremities | 51 | 23.9 |
| Number of lesions | 1–2 | 107 | 50.2 |
| | 3–4 | 89 | 41.8 |
| | ≥5 | 17 | 8 |

## Clinical characteristics of the lesions

As shown in Table 1, more than half, or 55.9% (n = 119) of the study participants had lesions on their heads, faces, and necks, 20.2% (n = 43) had lesions on their upper extremities, and 23.9% (n = 51) had lesions on their lower extremities. Around half of participants (50.2%) reported having 1–2 lesions and 41.8% of them had 3–4 lesions while others had five or more lesions. The average number of lesions participants reported was 1.61. Lesions lasted between 6 and 12 months in 42.7% of cases (n = 91), whereas more than half (57.3%) had lesions lasting less than 6 months. The mean duration of disease chronicity was 7.2 ± 4.1 months.

## Anxiety and depression among study participants

Figure 2 indicates that a significant proportion of the study participants experienced mild or moderate depression, with 55.9% (n = 119) falling into this category.

Similarly, Fig. 3 provides insights into the levels of anxiety among the female study participants in Hubuna. Among the participants, 27.2% (n = 58) exhibited a moderate level of anxiety, while 40.8% (n = 87) reported experiencing mild anxiety.

## Factors affecting anxiety and depression among participants

Based on the univariate analysis results presented in Table 2, age (OR = 1.012; 95% CI [0.894–1.009]) and location of CL lesions (OR = 1.414; 95% CI [0.73–2.16]) were not

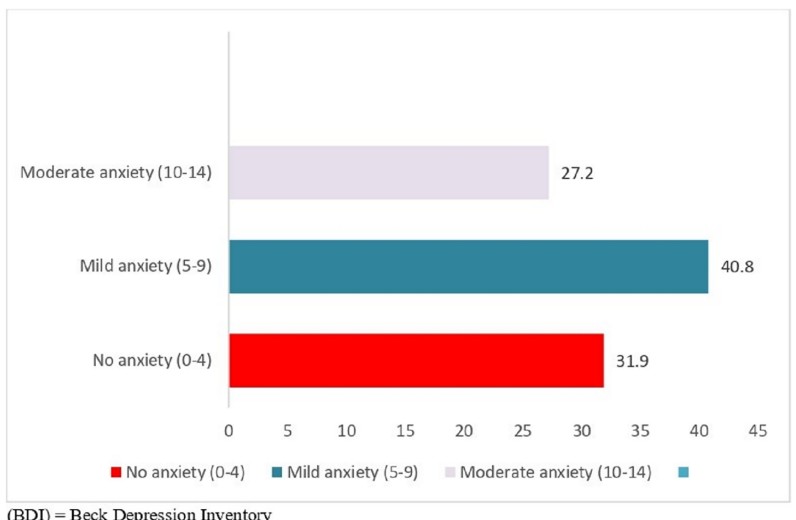

(BDI) = Beck Depression Inventory

**Figure 2 Levels of depression based on BDI scores among the sample (*n* = 213).** This figure shows the distribution of depression severity among the study sample (*n* = 213), as measured by the Beck Depression Inventory (BDI) scores. The x-axis displays the BDI score ranges, while the y-axis represents the frequency of participants falling within each score range. The bars are color-coded to indicate the severity of depression.                 

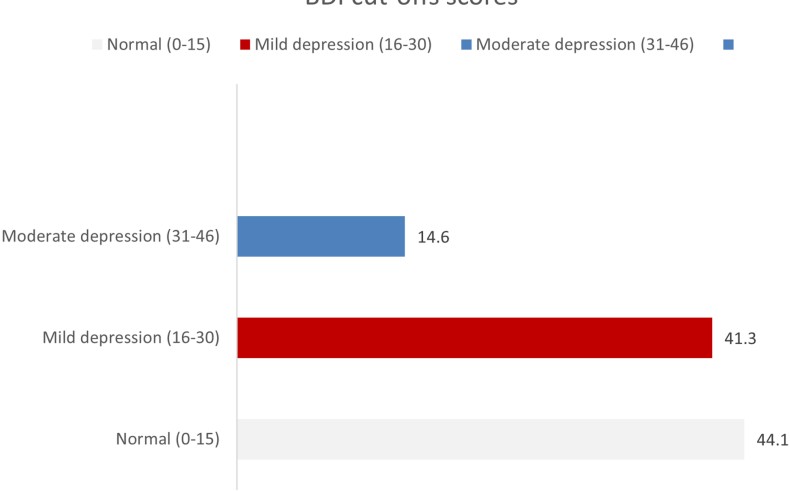

**Figure 3 Levels of anxiety based on GAD-7 scoring system among sample (*n* = 213).** This figure shows the distribution of anxiety severity among the study sample (*n* = 213), as measured by the Generalized Anxiety Disorder 7-item scale (GAD-7) scores. The x-axis displays the GAD-7 score ranges, while the y-axis represents the frequency of participants falling within each score range. The bars are color-coded to indicate the severity of anxiety.                 

significantly associated with depression among participants. Educational level, living environment, and employment status were also not significantly associated with depression (*p*-value ≥ 0.05). However, married women (OR = 1.059; 95% CI [0.761–1.404]) and the number of reported lesions (OR = 1.130; 95% CI [1.120–0.819]) were significantly associated with depression among participants, with *p*-values < 0.05. In case of 3–4 lesions, the (OR = 0.885; 95% CI [0.967–0.984]) suggests that participants

**Table 2 Univariate and bivariate logistic regression analysis of depression in regard to participants' characteristics (n = 213).**

| Variables | Prevalence n (%) | Simple logistic regression | | Multiple logistic regression | |
|---|---|---|---|---|---|
| | | OR (95 CI) | p-value | OR (95 CI) | p-value |
| Age (years) mean ± SD | 26.4 ± 14.8 | 1.012 [0.894–1.009] | 0.041 | 2.34 [1.450–4.12] | 0.001 |
| Marital status | | | | | |
| Single (did not experience marriage) | 116 (54.5) | Ref | | Ref | |
| Single (separated, divorced, widow) | 36 (16.9) | 1.059 [0.761–1.404] | 0.011 | 3.31 [2.87–3.79] | 0.002 |
| Married | 61 (28.6) | 1.187 [0.836–2.219] | 0.046 | 2.17 [1.87–3.52] | 0.065 |
| Level of education | | | | | |
| Uneducated or primary school | 41 (19.2) | Ref | | Ref | |
| Intermediate + secondary schools | 103 (48.4) | 1.008 [0.726–1.399] | 0.087 | 1.65 [1.34–2.64] | 0.076 |
| Higher education (University Graduates) | 69 (32.4) | 0.942 [0.668–1.328] | 0.061 | 1.65 [1.82–2.67] | 0.073 |
| Living environment | | | | | |
| Inside or nearby farms/multi-planted houses | 145 (68.1) | Ref | | Ref | |
| Far from farms/no plants inside houses | 68 (31.9) | 0.615 [0.518–0.807] | 0.761 | 0.95 [1.34–2.49] | 0.43 |
| Employment status | | | | | |
| Children/students/no jobs | 136 (63.8) | Ref | | Ref | |
| Employed female | 77 (36.2) | 0.937 [0.811–1.202] | 0.431 | 1.04 [1.75–2.24] | 0.67 |
| Location of lesions | | | | | |
| Head, face, and neck | 119 (55.9) | 1.414 [0.73–2.16] | <0.000 | 2.37 [3.27–4.48] | 0.68 |
| Upper extremities | 43 (20.2) | 0.519 [0.311–1.283] | 0.661 | 1.67 [1.56–2.82] | 0.19 |
| Lower extremities | 51 (23.9) | Ref | | Ref | |
| Number of lesions | | | | | |
| 1–2 | 107 (50.2) | Ref | | Ref | |
| 3–4 | 89 (41.8) | 0.885 [0.967–0.984] | 0.033 | 1.67 [1.56–2.56] | 0.001 |
| ≥5 | 17 (8) | 1.130 [1.120–0.819] | 0.001 | 0.87 [0.16–1.62] | 0.002 |

**Note:**
CI, Confidence Interval; OR, Odds Ratio; Ref., reference category.

with less lesions have slightly lower odds of depression compared to the reference group. The p-value of 0.033* indicates a statistically significant association. The participants with ≥5 lesions to the reference group. The (OR = 1.130; 95% CI [1.120–0.819]) seems to have higher odds of depression compared to the reference group. The p-value of 0.001* suggests a statistically significant association.

The findings of bivariate analysis in Table 2 suggest that age, marital status (especially being single and separated/divorced/widowed), level of education, living environment, employment status, location of lesions (specifically head, face, and neck), and the number of lesions (particularly having 3–4 lesions) are associated with depression.

The bivariate analysis results, presented in Table 3, revealed significant associations between various factors and anxiety among the participants. Females aged (OR = 1.52; 95% CI [11.43–1.87], p-value = 0.031) were found to have a significantly higher likelihood of experiencing anxiety. Being married (OR = 4.02; 95% CI [3.17–2.89], p-value = 0.000) was also significantly associated with a higher prevalence of anxiety. Additionally, participants

**Table 3 Univariate and bivariate analysis of socio-demographic characteristics and anxiety related to CL (n = 213).**

| Variables | Prevalence n (%) | Simple logistic regression | | Multiple logistic regression | |
|---|---|---|---|---|---|
| | | ORc (95% CI) | p-value | ORa (95% CI) | p-value |
| **Age** | | | | | |
| **Age (years) mean ± SD** | 26.4 ± 14.8 | 1.52 [1.43–1.87] | 0.031 | 1.76 [1.32–2.03] | 0.002 |
| **Marital status** | | | | | |
| Single | 101/152 (66.4) | 3.99 [3.11–12.28] | 0.041 | 4.02 [3.17–2.89] | 0.000 |
| Married | 44/61 (72.1) | Ref | | Ref | |
| **Level of education** | | | | | |
| Uneducated or primary school | 27/41 (65.9) | 1.56 [1.32–3.11] | 0.618 | 0.67 [0.51–1.22] | 0.081 |
| Intermediate + secondary school | 77/103 (75.5) | 2.38 [1.67–3.62] | 0.422 | 0.53 [0.43–0.89] | 0.062 |
| Higher education (University Graduate) | 41/69 (59.4) | Ref | | Ref | |
| **Living environment** | | | | | |
| Inside or nearby farms/multi-planted houses | 114/145 (78.6) | 1.42 [1.08–1.58] | 0.111 | 1.74 [1.04–2.91] | 0.212 |
| Far from farms/no plants inside houses | 31/68 (45.6) | Ref | | Ref | |
| **Employment status** | | | | | |
| Children/students/no jobs | 99/136 (72.8) | 1.18 [0.71–1.63] | 0.104 | 1.693 [1.042–2.750] | 0.556 |
| Employed female | 46/77 (59.7) | Ref | | Ref | |
| **Location of lesions** | | | | | |
| Head, face &neck | 82/119 (68.9) | 3.15 [3.88–2.94] | 0.000 | 4.160 [3.801–1.374] | 0.001 |
| Upper extremities | 33/43 (76.7) | 0.51 [0.34–0.76] | 0.056 | 2.209 [1.656–2.189] | 0.065 |
| Lower extremities | 30/51 (58.8) | Ref | | Ref | |
| **Number of lesions** | | | | | |
| 1–2 | 79/107 (73.8) | Ref | | Ref | |
| 3–4 | 55/89 (61.8) | 2.56 [2.28–1.90] | 0.001 | 2.942 [2.668–1.318] | 0.011 |
| ≥5 | 11/17 (64.7) | 1.82 [1.70–0.96] | 0.032 | 1.107 [1.726–1.389] | 0.045 |

**Note:**
CI, Confidence Interval; ORc, Odds Ratio Crude; ORa, Odds Ratio Adjusted; Ref., reference category.

with lesions on the face (OR = 4.160; 95% CI [3.801–1.374], p-value = 0.001) and three to four reported lesions (OR = 2.942; 95% CI [2.668–1.318], p-value = 0.011) exhibited a significantly higher likelihood of experiencing anxiety. On the other hand, there were no significant associations between anxiety and factors such as educational level, living environment, and employment status (p-value ≥ 0.05).

## DISCUSSION

Our study has shed light on the psychological impact of cutaneous leishmaniasis (CL) among women in the Hubuna region of Saudi Arabia. Our findings indicate that the female population in this region is particularly vulnerable to psychological problems such as anxiety, and depression, due to the lasting effects of CL, such as atrophic scarring. Our results are consistent with findings from other studies that have reported a higher incidence of mental health problems among individuals with cutaneous leishmaniasis, particularly among female CL patients (*Bully & Elosua, 2011*; *Hamdam, 2020*; *Karimkhani*

*et al., 2016*). The psychological effects of CL can be severe and long-lasting, affecting an individual's mental health and overall quality of life. In particular, the visibility of CL scars on the face and other exposed body parts can lead to social isolation. This can be particularly distressing for females who may face additional social pressures and expectations related to physical appearance.

Our study found that a significant proportion of females with CL in the Hubuna area of Saudi Arabia experienced CL-related depression and anxiety, with estimated rates of 55.9% and 68.1%, respectively. These findings are consistent with the results of other studies, including *Devrimci-Ozguven et al. (2000)* and *Yanik et al. (2004)*, who reported high rates of psychological distress among individuals with CL. *Devrimci-Ozguven et al. (2000)* investigated the psychosocial impact of CL scars on Turkish patients and found that 63% reported depressive symptoms and 53% reported anxiety symptoms. Similarly, *Yanik et al. (2004)* conducted a study among Syrian refugees with CL and found that 78% of participants reported psychological distress, including depression and anxiety (*Devrimci-Ozguven et al., 2000*; *Yanik et al., 2004*).

Studies conducted in various countries, including Yemen, Afghanistan, Tunisia, and Morocco, have shown that young females affected by cutaneous leishmaniasis may experience severe psychological and social consequences of the disease (*Bennis et al., 2017*). These consequences can be so severe that it may hinder their ability to get married; in some cases, women with CL are not even be allowed to stay with their partners (*Al-Kamel, 2016b*; *Reithinger et al., 2005*; *Chahed et al., 2016*).

*Al-Kamel, (2016b)* conducted a study among Yemeni females with CL and found that the disease had a negative impact on their perceived physical appearance, social status, and marriage prospects. Similarly, *Reithinger et al. (2005)* reported that Afghan females with CL experienced significant social stigma, which affected their ability to get married and to participate in social activities (*Al-Kamel, 2016b*; *Reithinger et al., 2005*).

*Chahed et al. (2016)* conducted a study among Tunisian patients with CL and found that females experienced higher levels of anxiety and depression than males. *Bennis et al. (2017, 2018)* also reported high rates of psychological distress, including depression and anxiety, among Moroccan patients with CL, with females being more affected than males (*Chahed et al., 2016*; *Bennis et al., 2018, 2017*).

Previous studies have also shown that females affected by CL may experience significant social and psychological consequences that can negatively impact their lives and relationships (*Reithinger et al., 2005*; *Al-Kamel, 2016a*; *Bennis et al., 2018*).

Our findings show that CL and its associated scars significantly impact the lives of the females in our study, who expressed their concerns and demanded treatment for their problems. This study also revealed feelings of humiliation due to being considered unsuitable for marriage; study participants expressed their deep concerns about the impact of CL on their marriage prospects without being prompted by any marriage-related questions. Several participants, especially those who reported CL lesions on their faces, reported being socially excluded, which they viewed as "social death". Similar findings were also reported in Afghanistan, Brazil, Pakistan, and India (*Chaturvedi, Singh & Gupta, 2005*; *Kassi et al., 2008*; *Stewart & Brieger, 2009*; *Toledo et al., 2013*).

The psychological impact of CL was observed to be significant among females in Hubuna, which is consistent with some studies that have reported a larger impact of CL on females than on males (*Okwor & Uzonna, 2016*; *Simsek et al., 2008*; *Uzun et al., 2018*). It has been observed that unmarried women were more likely to report depression and anxiety in Hubuna, which is consistent with previous research suggesting CL scars have a deeper impact on women who remain unmarried (*Nilforoushzadeh et al., 2012*; *Ranawaka, Weerakoon & De Silva, 2014*; *Weigel & Armijos, 2001*).

One of the major strengths of the current study is its community-based approach, which allowed for a large and diverse sample size. However, it is important to note that our study also has some limitations. The use of a self-administered questionnaire may have yielded less detailed information than individual in-depth consultations, limiting our ability to fully understand the underlying reasons behind the psychological symptoms induced by CL. Additionally, the experiences of the participants with CL lesions were based solely on self-declarations, which were not formally verified. Therefore, a complete physical examination would be needed to obtain more detailed information on the number and location of lesions.

## CONCLUSION

This study demonstrates that cutaneous leishmaniasis (CL) has a considerable impact on the psychological well-being of females in the Hubuna area, leading to high levels of anxiety and depression. The study identifies females as being significantly susceptible to the condition, emphasizing the need to develop targeted health education initiatives for this specific demographic. Such programs should focus on the prevention and management of CL and include measures to repel or eliminate the sandflies that transmit the disease. Overall, these findings underscore the need for targeted efforts to address the mental health impacts of CL in affected communities. It is important for healthcare providers to recognize the broader social and cultural contexts in which CL is experienced in order for them to provide appropriate support to affected individuals and it is important to consider treating the cases by approved cryotherapy and topical nitric oxide to minimize the psychological impact of CL.

### Funding

The project was funded by the Ministry of Education and the Deanship of Scientific Research at Najran University-Saudi Arabia under code number (NU/RG/MRC/11/1). The funders had no role in study design, data collection and analysis, decision to publish, or preparation of the manuscript.

### Grant Disclosures

The following grant information was disclosed by the authors:
Ministry of Education and the Deanship of Scientific Research at Najran University-Saudi Arabia: NU/RG/MRC/11/1.

## Competing Interests

Waled AM Ahmed and Wael Alghamdi are employed by Albaha University, Faroq Abdulghani Alshameri is employed by Alryan College, and all other authors employed by Najran University. We all declare there are no competing interests related to this study.

## Author Contributions

- Nahid Elfaki conceived and designed the experiments, performed the experiments, analyzed the data, authored or reviewed drafts of the article, and approved the final draft.
- Mohammed Alzahrani performed the experiments, analyzed the data, authored or reviewed drafts of the article, and approved the final draft.
- Yahya Hussein Ahmed Abdalla conceived and designed the experiments, authored or reviewed drafts of the article, and approved the final draft.
- Mugahed Ali Alkhadher conceived and designed the experiments, prepared figures and/or tables, authored or reviewed drafts of the article, and approved the final draft.
- Abdalla MohamedAhmed Osman performed the experiments, prepared figures and/or tables, authored or reviewed drafts of the article, and approved the final draft.
- Wargaa Taha analyzed the data, authored or reviewed drafts of the article, and approved the final draft.
- Wael Alghamdi conceived and designed the experiments, analyzed the data, prepared figures and/or tables, and approved the final draft.
- Faroq Abdulghani Alshameri conceived and designed the experiments, performed the experiments, prepared figures and/or tables, authored or reviewed drafts of the article, and approved the final draft.
- Waled AM Ahmed conceived and designed the experiments, performed the experiments, analyzed the data, prepared figures and/or tables, authored or reviewed drafts of the article, and approved the final draft.

## Ethics

The following information was supplied relating to ethical approvals (*i.e.*, approving body and any reference numbers):

The Deanship of Scientific Research at Najran University-Saudi Arabia approved the study (NU/RG/MRC/11/1).

## Data Availability

The raw data are available int the Supplemental File.

## Supplemental Information

Supplemental information for this article can be found online at http://dx.doi.org/10.7717/peerj.15582#supplemental-information.

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
