# Peer review of "Self-reported depression and anxiety rates among females with cutaneous leishmaniasis in Hubuna, Saudi Arabia"

_PeerJ, doi:10.7717/peerj.15582_

## Round 0.1 · original submission · Major Revisions

As commented by reviewer 1, including young children in the study is problematic because BDI and GAD-7 have not been validated in young children. Please confirm and address this as well as multiple other issues raised by the reviewers.

Reviewer 1 ·

Basic reporting

The manuscript contains some spelling errrors
The references needs to be updated. Most of references are related to studies published before the year 2015 and the most recent (1 reference) is published in 2020.
The introduction needs to be developped by describing the importance of CL n Saudi Arabia. Is it really important ?

Experimental design

One of the most important limitation of the study is the fact that the authors ncluded females from 4 to 61 years old. I have here muliple questions:
1. you staed in the title that you stadied women. Cn we consider a female aged 15 years or less (4 years??) as a woman? they are females
2. How did you evaluate the level of anxiety and stress among oung females?
3. Comparing the level of education by including females of 4 years seems to be not logical
3. How did you evaluate the level of anxiey among this category?

Validity of the findings

The discussion should be updated
The conclusion should be revised by descring the most important results (andrecommendation) (the same with the conclusion of the abstract

Additional comments

- Title:: women should be females
- Abstract: you should add the period of the study in the methds and add the descriptie results of your sample, modify the conclusion (as suggested before) and then summurize it.
Line 21: " CL was considerably higher among females in the Habouna area". First: here you studied only females and you can not concludethis. Second: this is not the conclusion of your study" Your study has not estimated the prevalence of CL?
Key words should not be in italic.
Introduction:
Line 30-31: "As....protozoan paasite". add the name of te parsite please
line 50-51: ".To our knowledge, some affected women often experience stressed marriages, and some may encounter many difficulties when getting married". Avoid to use self statements. you should developp more your adea s and add a rference.
Material and methods:
-You should describe the importance of CL in the study aera. This is the most important
Study design and sampling:
- Add the study period please
How did you calculate the number of 198. Explain more
- Line 83: which nationalities were included among he partcipants (explain this in the results).
- Line 87: "ta answered" correct please.
Data collecion:
- How did you distibuted through social media since any one could unswer to the questionnaire (because it is online). Your method is not clear. How did you conducted exactely the survey? were you guided? How did select the sample? Explain please
Statistcal analysis:
- I did not find where did you used t-test and ANOVA!!!!
Results:
- Line 130: 309 indiiduals. How did you do your survey with conactig them? explin in he methods please
- Line 146: Table two may be Table 2
Lines 147-153: you stated that the OR is significant but it has a CI including 0 (i.e CI0.894-1.009 mean that the result is not signifcant). Please correct all your results in this paragarh (including tables).
Discussion (to be upfated):
line 180:"The psychological impact of CL was observed to be greater among females than males in Habouna" .....
How did you find these reslts ??? you haen't estimated the prevalence of the disease in Harbouna!!! Aso, your study was conducted among females only!!!!Be careful
Line 185- 186: Are these your results? from where did you extract these results????
Conclusion:
".....and is considered an endemic health problem in Habouna". Are these your results? revise please as suggested before.
The same remark for lines 201-204: recommedation should be in relation with your work.
References:
References should be unified
Table 1: delere % from all variables (and in all the other tables)
delete the line after educational level
Table 2 and 4 will be better as figures with horizontal bars
Table 3 should be revised as suggested before)
Figure 1: the most imporant part of the methods is missed (how did you obtain the 309 individuals??) add more details please

·

Basic reporting

The manuscript is written in Clear and concise style.
The Intro & background showed the context.
The literature is well referenced & relevant.
The figure is relevant and well described.
The raw data was supplied, accessed and checked.

Experimental design

The paper reported original primary research within Scope of the journal.
The research question was well defined, relevant and meaningful.
However, it is not stated how the research fills an identified knowledge gap.
The investigation was performed with the respect of ethical standard. The technical used are not high.
The methods were described with sufficient detail and information to replicate.

Validity of the findings

All underlying data have been provided; they are robust, statistically sound, & controlled.
Conclusions are well stated, linked to original research question & limited to supporting results

Additional comments

The unknown research question is not stated clearly.
Abstract: line 6: the aim of this study was...
methods: 213 is the number of recruted patients, it is a result more than methodes (the requested number of 198 should be in the methods section.
results are not detailed: study population description

Main text: figure 1 should be in the results section
the discussion is weak and short (the introduction is longer than the discussion).

---

## Round 0.2 · Major Revisions

Please address additional issues raised by the reviewer, which may help improve the manuscript.

Reviewer 1 ·

Basic reporting

I would like to thank the authors for their efforts to improve the quality of the manuscript. However, I still have some comments:
The introduction needs to be reorganized again.
Lines 85–88: "Another... information.". This paragraph may be replaced in line 81 (the same idea).
Line 90: Of those affected by it delete please
Lines 95–99: Please separate the two ideas (psychological and social consequences). Social consequences were reported in line 85.
Lines 102–104: delete, please.

Experimental design

Line 112: Try to add more information about CL in Harbouna if available, please.
Line 127: Please explain which parameter you used to obtain this number (198). which assuption?
Line 178: write p value in lower case please (in all the text and in the tables)

Validity of the findings

.

Additional comments

Line 186: correct the sentence please
"had never been married" replace by singles please
Line 193: delete "This study included 213 female participants".
in the same line associate the sentence related to age with sentence of line 186.
Line 194: add "the" before "lesion".
Line 195: Add "the" .....of "the" study participants
Line 196: replace reporting by "reported"
Line 197: correct lesions (2 times)
Line 200: ..more than half.." add the percentage please
Line 202: add a subtitle for this part
Line 202-204: explain more and avoid to use the same sentences (figure show...)
Line 205: add a subtitle for this paragraph please
Line 205-207: .."age...included 0". delete please (you bigin with non significant factor, then significant factors and later you return to non significant factors).
Line 212-214: try to reformulate please (why did you chose only OR of one category for age, and the number of lesions. revise please and reformulate your sentences (use sentence like, females aged.., being married,..having lesion...) you should standardize
the more reported lesions???? ( write as: "having more than 2 lesions" for example
Here I have a real concern: why did you used only univariate analysis for depression and univariate and bivariate analyses for anxiety?? you should unify
Table 2: Correct "Frequency"
separate "(mild& moderate"
Table 5 may be Table 3
The title should be univariate and bivariate analysis
age: 10-20: coorect: 1.48 (1.31 – 1.77) add a bracket please
Figures 2 and 3 should have the same format
You should to use numbers or % not the two in the same figure (for the two figures)
delete severe depression and severe anxiety from the two figures

---

## Round 0.3 · Major Revisions

Please consider the reviewers' comments seriously and while you revise data analysis part, please also try to consult with a statistician or senior researcher.

Reviewer 1 ·

Basic reporting

see below

Experimental design

See below

Validity of the findings

see below

Additional comments

I would like to thank the authors for their efforts. However, the manuscript contains (again) multiple flaws that should be addressed before acceptance. Also, some of my comments were not taken into consideration or they did not take them seriously. The lack of a certain rigor and the lack of haste are apparent. The authors should take things more seriously and take their time to completely revise their manuscript.
1. The authors did not explain how did obtain he sample size of 198. They did add the statistical formula that they used to obtain this number. If it is not possible, they can delete this number (and surely the idea of the minimum sample size).
2. In the results, the authors used only univariate analysis for depression and univariate and bivariate analysis for anxiety. I would ask them to add the bivariate analysis for the depression in both the results and Table 2 (I asked you to do it but your response was not clear).
Also, you used different categories for the two parameter (anxietey and depression). Age for example is quantitative for depression but semi-quantitative for anxiety.
3. In the introduction, if the authors unswer to my request, they just move the paragraphs (or sentences) without effort to rely on the previous paragraphs and the order of ideas. The autours should use adapted adverbs according to the order of ideas (an example is in line 80, is additionnaly adapted here?. Revise the order of your ideas in the introduction, please.
I have also other concerns:
In the title, Hubana should be Hubuna (It is written like this in all the manuscript).
Line 210: " This finding highlights the considerable prevalence of depression among the female study participants" delete this sentence please (it should be in the discussion)
The same remark for line 214
Line 219: " as the confidence interval included 0 " delete, please.
Line 222: you should precise exactly how did the number of lesion affected depression. You added just the OR for those who have more than 5 lesions. explain please. The same is true for line 225, where you selected just the OR of one category.
Line 239: stigmatization was not among your results? Have you asked the participants if they were object to stigmatization?
Line 251: "...63% of patients"; delete patients here since it was cited before (Turkish patints).
Line 250-255 repeated nearly word by word the paragraph of line 96-100
Line 301: "The study identifies that females as being significantly... "(correct the sentence.)
Table 3: Correct the title please (there are 2 different titles).
Figure 2 and 3: delete the total, improve the quality of the figures and correct the titiles ( there are two different titles for each figure)

---

## Round 0.4 · Minor Revisions

There seems to be an objective error in the results reported in Table 2, for the multiple logistic regression analysis, living environment lacks a reference group. Please all carefully confirm if all other results are correct. Furthermore, for both the univariate and multiple logistic regression, the selection of reference group should be the same for both depression and anxiety, unless there are special reasons. Please carefully address this issue and accordingly revise the manuscript.

---

## Round 0.5 · accepted · Accept

Thank you for addressing all the concerns raised by the reviewers and editor.